# Identification of Immune Infiltration and the Potential Biomarkers in Diabetic Peripheral Neuropathy through Bioinformatics and Machine Learning Methods

**DOI:** 10.3390/biom13010039

**Published:** 2022-12-26

**Authors:** Wenqing Li, Jiahe Guo, Jing Chen, Haibo Yao, Renqun Mao, Chuyan Li, Guolei Zhang, Zhenbing Chen, Xiang Xu, Cheng Wang

**Affiliations:** 1Department of Hand and Foot Surgery, Huazhong University of Science and Technology Union Shenzhen Hospital, Shenzhen 518052, China; 2Department of Hand Surgery, Union Hospital, Tongji Medical College, Huazhong University of Science and Technology, Wuhan 430032, China

**Keywords:** diabetic peripheral neuropathy, immune cells infiltration, biomarkers, bioinformatics analysis

## Abstract

Diabetic peripheral neuropathy (DPN) is one of the most common chronic complications in diabetes. Previous studies have shown that chronic neuroinflammation was associated with DPN. However, further research is needed to investigate the exact immune molecular mechanism underlying the pathogenesis of DPN. Expression profiles were downloaded from the Gene Expression Omnibus (GEO) database. Differentially expressed genes (DEGs) were screened by R software. After functional enrichment analysis of DEGs, a protein–protein interaction (PPI) network analysis was performed. The CIBERSORT algorithm was used to evaluate the infiltration of immune cells in DPN. Next, the least absolute shrinkage and selection operator (LASSO) logistic regression and support vector machine-recursive feature elimination (SVM-RFE) algorithms were applied to identify potential DPN diagnostic markers. Finally, the results were further validated by qRT-PCR. A total of 1308 DEGs were screened in this study. Enrichment analysis identified that DEGs were significantly enriched in immune-related biological functions and pathways. Immune cell infiltration analysis found that M1 and M2 macrophages, monocytes, resting mast cells, resting CD4 memory T cells and follicular helper T cells were involved in the development of DPN. LTBP2 and GPNMB were identified as diagnostic markers of DPN. qRT-PCR results showed that 15 mRNAs, including LTBP2 and GPNMB, were differentially expressed, consistent with the microarray results. In conclusion, LTBP2 and GPNMB can be used as novel candidate molecular diagnostic markers for DPN. Furthermore, the infiltration of immune cells plays an important role in the progression of DPN.

## 1. Introduction

According to the tenth edition of IDF Diabetes Atlas 2021, 537 million people are suffering from diabetes, and this number is projected to be 783 million by 2045 [1]. DPN is one of the most prevalent chronic complications and the cause of limb amputations in diabetes mellitus (DM) [2]. Pain and numbness are typical and serious symptoms of patients with diabetic peripheral neuropathy (DPN). However, it shows no obvious clinical symptoms or manifestations in the inchoate stages. At present, the gold standard methods for diagnosing DPN are usually based on the electroneuromyography examination [3]. In practice, these diagnostic methods are difficult and impractical to implement as they are time-consuming and labor-intensive. Thus, there is still a lack of precise early diagnostic indicators of DPN. To improve the quality of life of patients with DPN, prevention by tight glucose control and lifestyle intervention is the best current treatment for DPN. Therefore, the biomarkers for early diagnosis are critical in improving the early diagnosis of DPN patients, which may also improve the prognosis of DPN.

Previous investigations have demonstrated that both ischemic and metabolic factors play a key role in DPN [4]. Among those mechanisms in DPN, oxidative stress and chronic neuroinflammation have been highlighted by multiple reviews and research articles [5,6,7]. However, the multifactorial and complex pathogenetic mechanisms in DPN have not yet been fully elucidated. To further explore the specific molecular mechanism, transcriptomics analyses have been performed in several studies, most of them utilizing the gene microarray [8]. Microarrays are commonly used in performing gene expression studies to clarify the relationship between multiple different genes and the disease. In a recent study, a microarray was performed on the sciatic nerve tissues from control rats and DPN rats. The results identified a pool of candidate biomarkers involved in the early phase of experimental DPN [9]. Another study found that the neurotrophin-MAPK signaling pathway was a key signaling pathway in the development of DPN [10]. Based on these previous studies, differentially expressed genes (DEGs) were identified in our study using published datasets in the GEO database which contains DPN and normal sciatic nerve samples.

Biomarkers can provide accurate early diagnosis and guidance in clinical decision-making. Additionally, they have contributed to the objective evaluation of pathogenic processes. The neurophysiological methods found that some electrophysiological indicators are expected to be widely used as diagnostic and predictive biomarkers [11]. However, traditional biomarkers have shortcomings and limitations, and few have been used clinically [12]. Recently, molecules involved in several metabolic and signaling pathways associated with DPN have been suggested as predictive biomarkers [13].

A growing number of studies have revealed that neuroinflammation serves an important function in the occurrence and development of DPN [14]. For example, dorsal root ganglia are infiltrated by T-cells and neutrophils in chronic DPN [15]. Therefore, from the perspective of the immune system, evaluating the infiltration of immune cells in peripheral nerves and determining the differences in immune cell infiltrate composition would be valuable for elucidating the molecular mechanisms of DPN and developing new immunotherapeutic targets. CIBERSORT is an algorithm used to evaluate gene expression data from microarrays and analyze various immune cell proportions inside the samples [16]. It has been extensively used in the immune cell infiltration analysis in many diseases such as rheumatoid arthritis, lupus nephritis, idiopathic pulmonary fibrosis and human cancers [17,18,19]. To date, no prior studies have yet analyzed immune cell infiltration using CIBERSORT in DPN.

In the present study, we obtained gene expression microarray data of DPN from the GEO database. Then, freely available and open-source bioinformatic tools were used to identify differentially expressed genes in DPN samples and normal samples. Functional and pathway enrichment analysis and protein–protein interaction (PPI) network analyses were conducted. We aimed to unravel the specific molecular mechanisms by which these DEGs contribute to the development and progression of DPN. Next, the CIBERSORT algorithm was applied to analyze the difference in immune infiltration between DPN and normal nerve tissues for the first time. Subsequently, machine learning algorithms were used to further screen and determine the potential biomarkers of DPN. Finally, to further understand the immune mechanisms during DPN development, the relationship between the biomarkers and the infiltrating immune cells was studied. In addition, 15 mRNAs were confirmed as differentially expressed by qRT-PCR. The complete workflow is shown in Appendix A.

## 2. Materials and Methods

### 2.1. Data Preprocessing and DEGs Identification

We searched microarray gene expression profiling of diabetic peripheral neuropathy from the Gene Expression Omnibus (GEO, https://www.ncbi.nlm.nih.gov/geo/, accessed on 10 November 2022) database. Dataset GSE70852 [20] and GSE27382 [21] were selected for further analysis. GSE70852 contains microarray measurements of dorsal root ganglia (DRG) and sciatic nerve (SCN) tissue from 26-week-old ob/+ and ob/ob mice (*n* = 5 in each group). The GSE27382 dataset contains 6 samples from 24-week-old BKS db/db mouse sciatic nerve and 7 samples from db/+ mouse sciatic nerves. We chose SCN samples in two datasets (*n* = 23) for further analysis. Then, GSE70852 and GSE27382 gene expression matrices were merged, and the batch effect was removed using the “sva” package of R software [22] (version 4.1.2, http://r-project.org/, accessed on 10 March 2022). The effect of removing batch effects was demonstrated using a box plot and a two-dimensional PCA cluster plot was used to evaluate the effect of inter-sample correction.

### 2.2. Differential Expressed Genes Screening and Analysis

The R package “limma” was applied to the normalized and merged gene expression matrix to identify significantly DEGs [23]. DEGs with *p* < 0.05 and |Foldchange| > 1.5 were considered statistically significant [24]. The R package “pheatmap” (https://CRAN.R-project.org/package=pheatmap, accessed on 10 March 2022) was used to construct the heatmap plot. A volcano plot was constructed using the OmicStudio tool at https://www.omicstudio.cn/tool (accessed on 10 November 2022).

### 2.3. Differential Expressed Genes Screening and Analysis

Gene Ontology (GO), Kyoto Encyclopedia of Genes and Genomes (KEGG) functional enrichment analyses and gene set enrichment analysis (GSEA) were conducted based on the DEGs using the R package “clusterProfiler” [25]. The R package “GOplot” and “enrichplot” were used to visualize the results of enrichment analysis [26]. Disease Ontology (DO) enrichment analysis was performed on DEGs through the DisGeNET database [27]. A false discovery rate (FDR) < 0.05 and a *p* < 0.05 were considered significant enrichments.

### 2.4. Construction of the PPI Network of Differential Expressed Genes and Hub Genes Analysis

To explore the relationship of DEGs, a PPI network was constructed using the STRING online database (version 11.5, https://www.string-db.org, accessed on 10 March 2022) [28], with interactions with a combined score > 0.9 being used for network construction. Cytoscape v3.8.1 was used to visualize the PPI network. The cytoHubba plugin in Cytoscape was employed to identify hub genes based upon eight algorithms, including stress, radiality, MNC (maximum neighborhood component), MCC (maximal clique centrality), EPC (edge percolated component), EcCentricity, DMNC (density of maximum neighborhood component), degree, closeness, BottleNeck and betweenness [29]. We selected the top 70 node genes scored by each algorithm to screen hub genes in DPN. An UpSet plot was generated using the R package “UpSetR” [30].

### 2.5. Evaluation of Immune Cell Subtype Infiltration

The abundance of 22 types of infiltrating immune cells of each sample with DPN or normal was estimated by translating the gene expression matrix data into the relative proportion of immune cells [16]. The 22 types of immune cells include naive B cells, memory B cells, plasma cells, CD4+ T cells, CD8+ T cells, resting and activated NK cells, monocytes, M0/M1/M2 macrophages, dendritic cells, mast cells, eosinophils, etc. This was achieved with the CIBERSORT algorithm based on deconvolution using the R package “CIBERSORT” (http://cibersort.stanford.edu/, accessed on 20 March 2022). Analysis was performed by using the default signature matrix at 1000 permutations [31]. Then, PCA clustering analysis was performed and a correlation heatmap was drawn by the OmicStudio tool. R packages “ggplot2” and “ggpubr” were applied to visualize the results from CIBERSORT [32]. Here, only the data with *p* < 0.05 were used for subsequent analysis. The least absolute shrinkage and selection operator (LASSO) logistic regression model was conducted to analyze the different infiltrates of immune cells in DPN and normal samples with the R package “glmnet” [33].

### 2.6. Identification and Verification of Biomarkers

LASSO logistic regression and the support vector machine-recursive feature elimination (SVM-RFE) machine learning method were applied to identify the potential biomarkers associated with DPN [34,35]. RFE-SVM was implemented with the R package “e1071” (https://cran.r-project.org/web/packages/e1071/index.html, accessed on 11 April 2022). The RNA-Seq dataset GSE159059, which contains 10 non-diabetic db/+ mice and 10 DPN mice, was used as the validation dataset [36]. The gene expression matrix of the validation RNA-Seq dataset was downloaded from the GEO database. After combining the DEGs selected by the LASSO and SVM-RFE algorithms, potential biomarkers were identified by the two algorithms simultaneously. The receiver operating characteristic (ROC) curve was applied to evaluate the diagnostic value of biomarkers.

### 2.7. Correlation Analysis between Diagnostic Markers and Immune Cells

Spearman correlation analyses were performed to assess the correlation between the diagnostic markers and infiltrating immune cells [37]. The results were visualized by R package “ggplot2”.

### 2.8. Animals

Male, 12-week-old, nondiabetic C57BL/ksJ-leprdb/lepr+ mice (db/+) and diabetic C57BL/ksJ-leprdb (db/db) mice (*n* = 5 per group) were purchased from Huafukang Company (Beijing, China). The animals were housed under standard conditions of a 12 h light/dark cycle and given unrestricted access to water and food. Mice were handled in accordance with the National Institutes of Health Guidelines and Regulations, and all experiments were approved by the Animal Ethics Committee of Huazhong University of Science and Technology.

### 2.9. Tissue Harvest and Quantitative Real-Time PCR

At 26 weeks of age, all the animals were killed by sodium pentobarbital overdose after random blood glucose level monitoring and behavioral tests were conducted [38]. Then, sciatic nerves from two groups were dissected and used for RNA extraction. Total RNAs were extracted using a QIAGEN RNeasy Mini Kit. qRT-PCR was performed to validate the expression level of DEGs based on the instruction of ChamQ SYBR qPCRMaster Mix (Vazyme, Nanjing, China). The RNA data were normalized to β-actin as the endogenous reference. Gene expression levels were calculated with relative expression levels by using delta–delta Ct method (2^−△△Ct^). The primer sequences were listed in Appendix A.

### 2.10. Statistical Analysis

Data were expressed as the mean ± SEM. *p* < 0.05 was considered statistically significant. The unpaired Student’s *t*-test and one-way analysis of variance with Bonferroni post hoc test were performed for comparisons between two groups. Statistical analysis was calculated with the GraphPad Prism v 9.3.1 software.

## 3. Results

### 3.1. Data Preprocessing and DEGs Identification

The microarray data of the GSE70852 and GSE27382 datasets were merged, containing 12 DPN sciatic nerve samples and 11 normal sciatic nerve samples. The box plot shows that the batch effects between two gene expression profile datasets were removed (Figure 1a,b). After normalization and batch effect removal, principal component analysis was used to characterize the merged dataset (Figure 1c,d). |Foldchange| > 1.5 and *p* < 0.05 were used as the thresholds to screen differentially expressed genes in DPN after data preprocessing. A total of 1308 DEGs were obtained from the gene expression matrix using the R package “limma”, including 628 upregulated and 680 downregulated genes (Appendix A). A heatmap and volcano map are shown in Figure 2.

### 3.2. Functional Enrichment and Pathway Analyses

To determine functions associated with DEGs in DPN, GO analysis was performed based on the 628 upregulated DEGs and 680 downregulated DEGs. A GO circle plot highlights the top 10 GO biological process (BP) terms that are strong candidates for DPN (Figure 3a,b). The inner ring of the circle plot represents a bar plot, where the bar height indicates the negative log *p* value of the BP term described. The outer ring shows a scatter plot of the expression levels of DPN associated DEGs in each enriched GO term [39]. GO analysis results showed that upregulated DEGs were mainly related to the biological activity of inflammatory cells, such as leukocyte migration and neutrophil migration (Figure 3a). Downregulated DEGs were mainly related to neural functions (Figure 3b). The above results suggested that the immune response plays an important role in DPN. KEGG pathway analysis and GSEA analysis were also used to reveal the changed biological pathways in DPN (Figure 3c,d,f). Overlapping the KEGG pathways analysis with the GSEA results produced three pathways, one of them was the IL-17 signaling pathway, which is also a key pathway in regulating immunity [40]. DisGeNET is a discovery platform integrating information on gene-disease associations from public data sources and the literature [27]. Furthermore, the diseases associated with DEGs were mapped to the DisGeNET database and the results are shown in Figure 3e.

### 3.3. PPI Network Analysis of DEGs

In order to identify potential links between DEGs, a PPI network with 381 nodes and 776 edges was generated using the STRING database. The confidence interaction score was set to 0.9 in Appendix A for the network construction. Then, eight algorithms in the cytoHubba plugin were used to calculate the score of DEGs. As a result, seven hub genes were screened by the R package “UpSetR” (Figure 4a). The expression levels of the seven hub genes in the merged dataset are shown by a heatmap (Figure 4b). These results suggested that these hub genes may play important roles in the development of DPN.

### 3.4. Immune Cell Infiltration in DPN and Normal Tissues

PCA cluster analysis results of immune cell infiltration showed that there was a significant difference in immune cell infiltration between the DPN samples and the control samples (Figure 5a). The correlation analysis results between different types of immune cells are represented by a correlation heatmap. It showed that monocytes had a significant positive correlation with M2 macrophages. Resting dendritic cells and M1 macrophages also had a positive correlation. On the other hand, resting dendritic cells had a significant negative correlation with monocytes and M2 macrophages (Figure 5b). The infiltrating levels of 22 immune cell types in each sample are presented in a histogram using the function of R package “RColorBrewer” (Figure 5c). The box plot of the immune cell infiltration difference shows eight types of immune cells with *p* < 0.05 (Figure 6a). Seven types of immune cells were selected by LASSO and the results are displayed in Figure 6b,c. Taking the intersection of the two methods, six types of immune cells were considered to be differentially expressed. Specifically, compared with the normal control sample, M1 macrophages and resting CD4 memory T cells infiltrated more in DPN samples, while M2 macrophages, resting mast cells, monocytes and follicular helper T cells infiltrated less.

### 3.5. Screening and Verification of Biomarkers Markers

The LASSO logistic regression algorithm and SVM-RFE algorithm were used to screen out novel diagnostic biomarkers most associated with DPN from DEGs. Ten and eight candidate genes were identified by LASSO and SVM-RFE algorithms, respectively (Figure 7a,b). Finally, two diagnostic related genes (LTBP2 and GPNMB) were obtained by taking the intersection of the two algorithms (Figure 7c). To make the validation more credible, further validation of the diagnostic efficacy of LTBP2 and GPNMB was performed in the validation dataset GSE159059. The expression levels of LTBP2 and GPNMB in the dataset GSE159059 are shown in Figure 7d,e. The ROC curve showed the diagnostic performance of LTBP2 and GPNMB in the verification dataset and the area under the ROC curve (AUC), which can summarize the overall diagnostic accuracy of the potential biomarkers, was 0.896 (Figure 7f), indicating that LTBP2 and GPNMB had high diagnostic value.

### 3.6. Correlation Analysis between LTBP2 and GPNMB, and Infiltrating Immune Cells

The correlation analysis showed that LTBP2 was positively correlated with M1 Macrophages and resting CD4 memory T cells, and it was negatively correlated with M2 macrophages, monocytes and follicular helper T cells (Figure 7g). GPNMB was positively correlated with resting CD4 memory T cells, and it was negatively correlated with M2 macrophages, monocytes, follicular helper T cells and resting mast cells (Figure 7h).

### 3.7. Validation of DEGs by qRT-PCR

To further validate the above outcomes obtained from microarray analysis, qRT-PCR was carried out. We selected 20 DEGs with high fold changes or high weight in the network to validate the analysis results. The total RNA from individual mouse sciatic nerves (*n* = 5 per group) was extracted and then evaluated for these genes. As a result, 15 DEGs have been verified. The gene expression levels of UBD, UCP1, LTBP2, CCL2, S100A8, HSPB7 and GPNMB were increased in the DPN groups compared with the control groups. Furthermore, MMP9, PON1, CYP2F2, CDH1, TUBB3, MYT1L, CACNB4 and MGL2 were significantly downregulated in the DPN groups (Figure 8). Among these DEGs is the inflammatory response-related gene CCL2, which is reported to be associated with diabetic neuropathic pain [41]. A study has found that S100A8 expression levels increased in neurodegenerative disorders and inflammatory and autoimmune diseases [42]. Uncoupling protein 1 (UCP1) is a 32-kDa protein located in the inner membrane of mitochondria. It regulates the dissipation of excess energy via uncoupling oxidative phosphorylation from ATP synthesis [43]. Recent investigations have suggested that differential regulation of UCPs may be associated with diabetes and DPN [44]. Paraoxonase 1 (PON1) has been extensively evaluated as a genetic candidate for diabetic microvascular complications [45]. TUBB3 is primarily expressed in neurons and may be involved in neurogenesis and axon guidance and maintenance. More importantly, there was a significant difference in the expression levels of two biomarkers, LTBP2 and GPNMB, between two groups.

## 4. Discussion

DPN is a common, serious and troublesome chronic complication of DM [46]. Chronic hyperglycemia and oxidative stress lead to major structural and functional abnormalities of the peripheral nerves. In addition, neuroinflammation plays an important role in the development of DPN. Currently, large numbers of DPN patients complain of pain, fatigue, reduced quality of life and disability. Unfortunately, early diagnosis is difficult due to the lack of specific diagnostic indicators. Therefore, finding novel diagnostic biomarkers and analyzing the pattern of DPN immune cell infiltration is useful for improving the outcomes of patients with DPN. Previously, multiple studies have found that the signaling pathways comprised of some genes may play an important role in the development of DPN. However, few systematic analyses and comparisons of the transcriptome data have been made. More importantly, the exact mechanism underlying the progression of DPN driven by key genes remains to be fully elucidated.

In this study, bioinformatics techniques were used to analyze microarray data acquired from the GEO database isolated from sciatic nerves of T2DM mouse models to identify potential biomarkers. A total of 628 upregulated and 680 downregulated DPN-related DEGs were identified in the GSE70852 dataset and GSE27382 dataset. GO enrichment analysis showed that upregulated DEGs were mainly enriched in leukocyte migration (GO:0050900), leukocyte chemotaxis (GO:0030595), cell chemotaxis (GO:0060326), neutrophil migration (GO:1990266) and granulocyte migration (GO:0097530). Additionally, downregulated DEGs were mainly enriched in neurotransmitter transport (GO:0006836), regulation of membrane potential (GO:0042391) and axonogenesis (GO:0007409). From the above results, it was revealed that a DPN upregulated immune response and was significantly associated with the impairment of neurological function. Furthermore, DO enrichment analysis showed that immune-mediated diseases such as inflammation, fibrosis and arthritis were enriched. The IL-17 signaling pathway, p53 signaling pathway and Toll-like receptor signaling pathway were identified to be associated with DEGs by pathway enrichment analysis. Several investigators have suggested that upregulated IL-17 possesses a crucial role in the inflammatory process and the development of DM [47]. Ben Y et al. found that astragaloside IV could reduce the occurrence of mitochondrial-dependent apoptosis by regulating the SIRT1/p53 pathway in DPN rats [48]. Other studies have found that Toll-like receptor4 could be a potentially sensitive diagnostic biomarker for DPN in type 2 diabetic patients [49]. Our analysis data were also consistent with the findings above.

Through PPI network construction, genes that have high scores in eight algorithms were considered as key hub genes, such as CCL2, TGFB1, MMP9 and CD68. It is of note that abnormal expression of some genes has been reported to be related to DM or DPN in the past few years. As an example, C-C chemokine ligand 2 (CCL2) and its receptor are key players in the attraction of monocytes to sites of injury and inflammation and it was proposed to be a major cause of diabetic neuropathic pain [41,50]. Previous studies have demonstrated that Triphala churna acted as a neuroprotective agent in DPN via the downregulation of inflammatory cytokines such as TGFB1 [51]. Moreover, downregulation of MMP9 could improve peripheral nerve function via promoting Schwann cell autophagy in DPN [52]. A previous study found that CD68, a macrophage marker, was higher in the DRGs of patients with DPN, demonstrating that the upregulated inflammatory markers may contribute to the inflammatory response, potentially stemming from diabetes related neuronal pathology [53]. Overall, inflammation can be an important factor following peripheral nerve injury, as activated macrophages are needed to engulf myelin debris and apoptotic cells. However, sustained and low-grade inflammation is generally known to be linked to diabetes [54]. This impairs the cell viability in the peripheral nerve.

In order to explore the role of immune cell infiltration in DPN, CIBERSORT analysis was applied to estimate the fractions of immune cells in sciatic nerves. We found that an increased infiltration of M1 macrophages and resting CD4 memory T cells, and a decreased infiltration of M2 macrophages, resting mast cells, monocytes and follicular helper T cells may be related to the development of DPN. Macrophages are professional phagocytes belonging to the innate immune system that can be activated by a variety of external stimuli. Based on their function, macrophages can be differentiated into two phenotypes: M1 (pro-inflammatory) and M2 (anti-inflammatory) macrophages [55]. M1 macrophages are able to secrete a broad range of inflammatory factors, such as IL-6, IL-1β and TNF-α. Previous studies have shown that M1 macrophages increased significantly in DPN patients [56]. This means that M1 macrophages might play a pivotal role in the onset and development of DPN [57]. Moreover, it was found that increased expression of TLR4 in monocytes could be related to systemic inflammation in peripheral neuropathy in T2DM [58]. These findings further support the important role of immune cell infiltration and inflammation in the development of DPN.

LASSO logistic regression is a reliable method for selecting diagnostic features of DPN based on regression trees. It provides a statistically rigorous method to identify the variable λ when the predicted outcomes are best. Furthermore, SVM-RFE is a classic machine learning method based on a recursive feature elimination strategy to select important genes by training a support vector machine model. To further select feature variables and build an accurate classification model, we applied these two algorithms in this study. The overlap of the LASSO logistic regression model and the SVM-RFE algorithm was obtained. Consequently, LTBP2 and GPNMB were recognized as potential diagnostic markers for DPN.

Latent transforming growth factor beta binding protein 2 (LTBP2) is a member of the fibrillin/LTBP extracellular matrix glycoprotein family [59]. It plays a critical role in regulating the extracellular matrix glycoprotein. A growing number of studies have found that LTBP2 was associated with cardiac fibrosis, acute heart failure, glomerular filtration rate and pre-eclampsia [59]. Recent investigations have suggested that overexpression of LTBP2 facilitated inflammation in endometriosis [60]. It is regretful that the role of LTBP2 in DPN development has not been studied. Therefore, this needs further experimental verification. GPNMB is an endogenous type 1 transmembrane glycoprotein. A study has shown that GPNMB is closely related to neuroinflammation [61]. Interestingly, neuroinflammation happens to be one of the most important mechanisms in the development of DPN. It would be reasonable to speculate that GPNMB may play an important role in the disease progression of DPN. In conclusion, evidence from previous studies indicates that LTBP2 and GPNMB may play an important role in the development and progression of DPN. However, validated experiments and clinical studies are still needed to assess the diagnostic value of LTBP2 and GPNMB. A comprehensive analysis was performed including LTBP2, GPNMB and immune cells. LTBP2 was significantly positively correlated with M1 macrophages and GPNMB was significantly negatively correlated with M2 macrophages. We speculate that LTBP2 and GPNMB affect immune cells to participate in the occurrence and progression of DPN. Further experimentation is needed to validate these hypotheses, including experiments regarding the interactions between genes and immune cells.

## 5. Conclusions

In this study, DEGs associated with DPN were identified by analyzing previously published datasets containing DPN and normal sciatic nerve samples. Then, functional enrichment and PPI network analyses were conducted for DEGs, elucidating the detailed mechanisms and the pathogenesis of DPN. What is more, by using novel bioinformatics methods such as LASSO logistic regression algorithms and the SVM-RFE algorithm, we have identified potential DPN diagnostic markers, LTBP2 and GPNMB. This is the first time that CIBERSORT was used to analyze immune cell infiltration in peripheral nerve tissues. Nevertheless, we recognize that there were important limitations in our study which cannot be ignored. First, the current study is limited by a small sample size due to the small number of gene microarrays in DPN. Furthermore, CIBERSORT analysis is based on limited genetic data that may deviate from heterotypic interactions of cells, disease-induced disorders or phenotypic plasticity. In addition, our research needs to be further experimentally validated. In conclusion, our results present the promising potential for several diagnostic biomarkers of DPN and provide a novel strategy for DPN diagnosis and treatment.

## Figures and Tables

**Figure 1 biomolecules-13-00039-f001:**
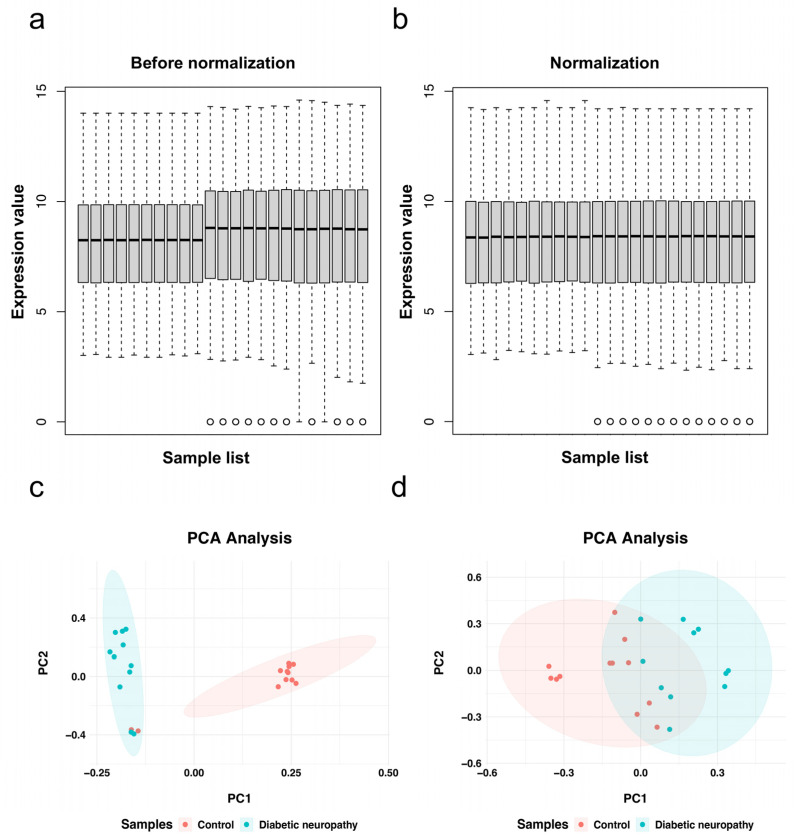
Data preprocessing. Box plot and principal component analyses were performed to remove the batch correction of merged datasets including GSE70852 and GSE27382. The box plots before and after data normalization (**a**,**b**). The PCA plots before and after batch correction (**c**,**d**).

**Figure 2 biomolecules-13-00039-f002:**
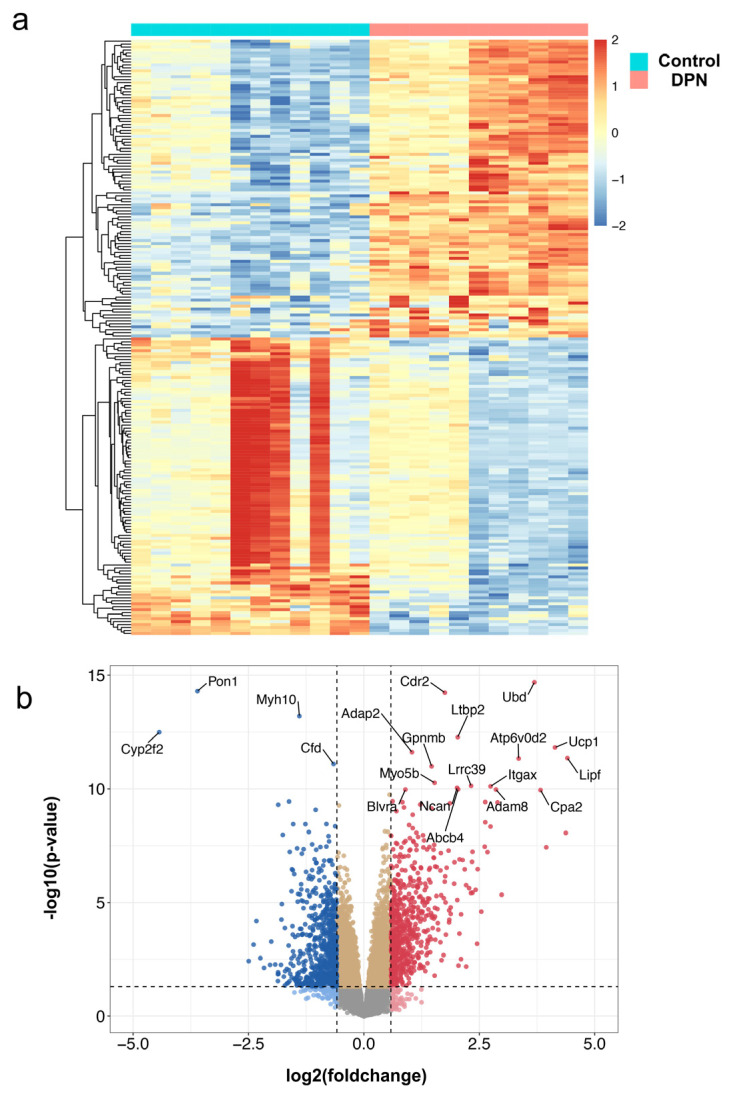
Identification of DEGs. (**a**) The heatmap of the DEGs derived from integrated analysis. The red and blue colors represent upregulated DEGs and downregulated DEGs, respectively. (**b**) The volcano plot of normalized microarray data. The upregulated and downregulated genes are marked in red and blue, respectively.

**Figure 3 biomolecules-13-00039-f003:**
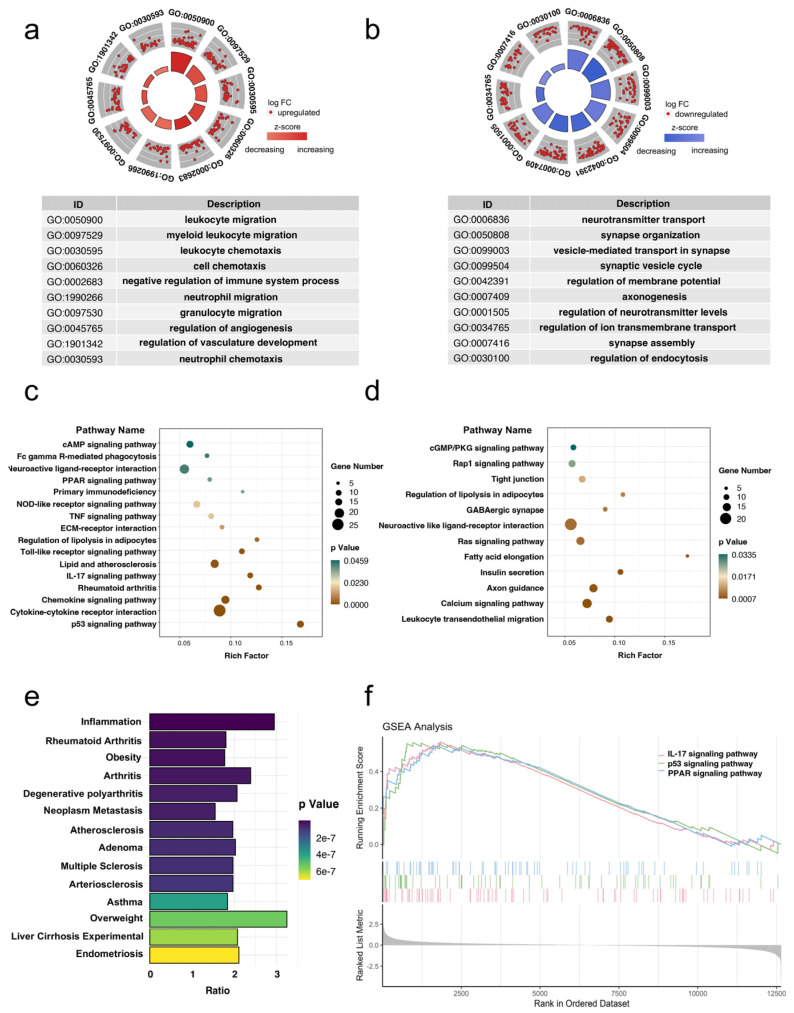
Enrichment analyses of gene ontology (GO), KEGG pathway, disease ontology (DO), and gene set enrichment analysis (GSEA). (**a**) The upregulated biological processes and (**b**) the downregulated biological processes in the DPN group. The top 10 are listed. The nodes in the concentric circle represent the DEGs clustered in the GO annotations. The red and blue colors represent upregulated DEGs and downregulated DEGs, respectively. (**c**,**d**) KEGG pathway enrichment analysis of upregulated and downregulated DEGs of the DPN group. (**e**) The enrichment analysis of disease ontology. (**f**) Gene set enrichment analysis (GSEA). The IL-17 signaling pathway, P53 signaling pathway and PPAR signaling pathway were significantly enriched.

**Figure 4 biomolecules-13-00039-f004:**
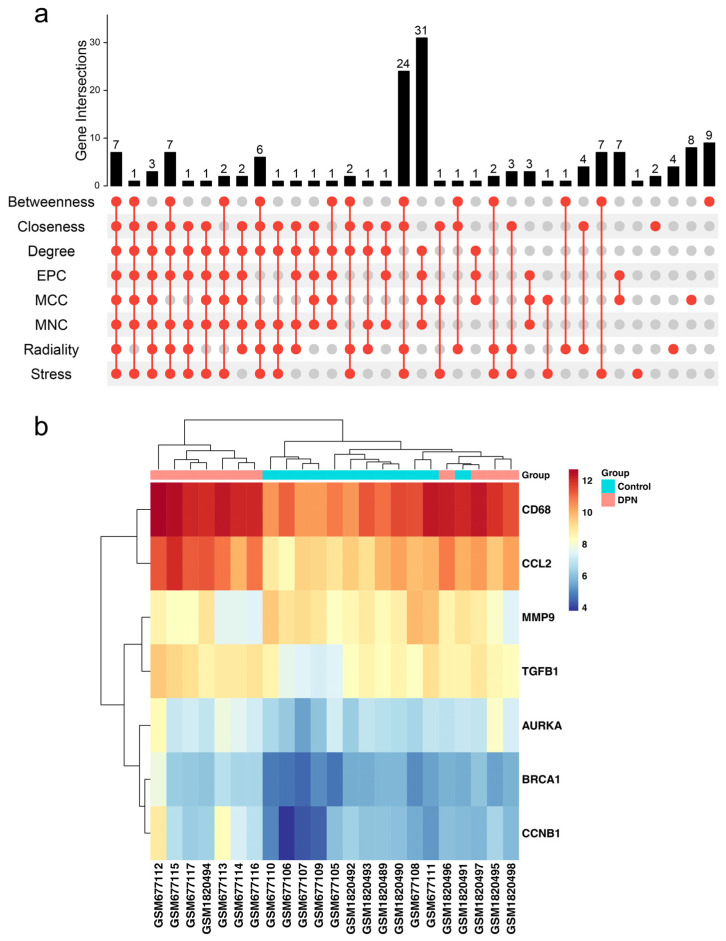
Identification of hub genes in the PPI network. (**a**) Eight algorithms were used to screen hub genes by the R package “UpSet”. (**b**) The expression of hub genes is presented by a heatmap in the merged microarray data.

**Figure 5 biomolecules-13-00039-f005:**
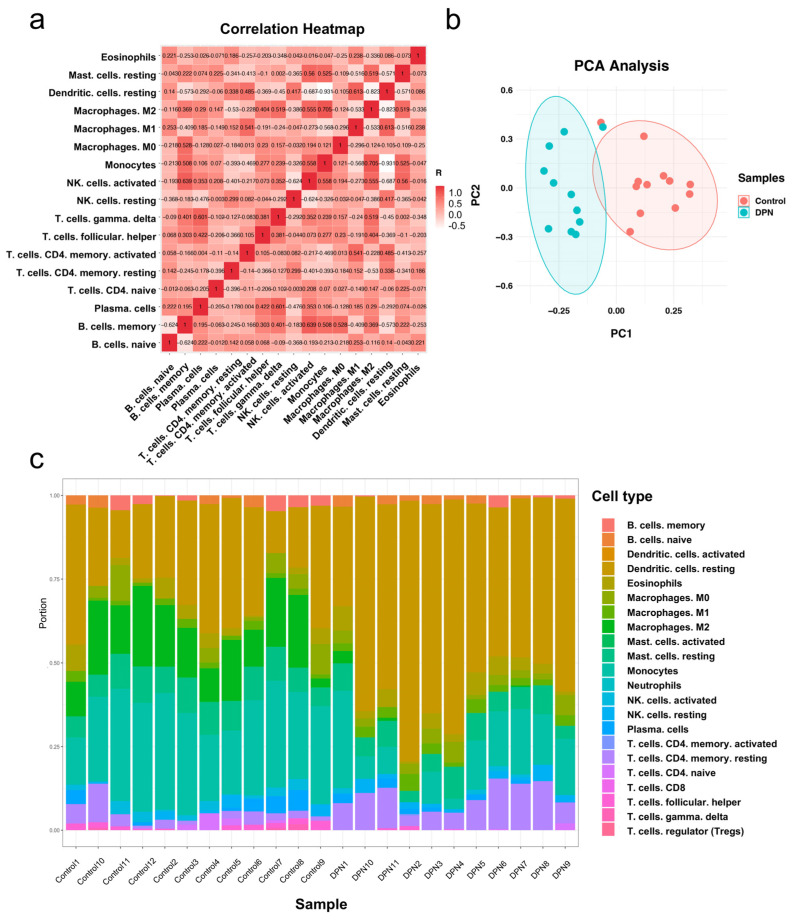
Evaluation and visualization of immune cell infiltration in DPN and normal samples. (**a**) The correlation of immune cells in DPN samples was evaluated. Different shades of squares represent the degree of negative or positive correlation. (**b**) Principal component analysis (PCA) cluster plot of immune cell infiltration between DPN and normal samples. (**c**) A histogram plot shows the composition of immune cells in each sample.

**Figure 6 biomolecules-13-00039-f006:**
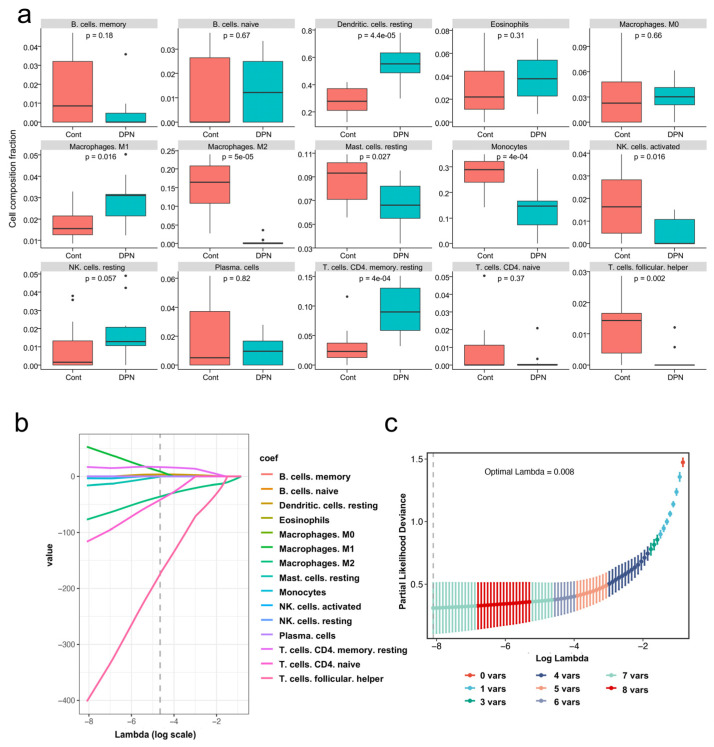
Identifying the infiltration levels of immune cell populations in DPN. (**a**) Box plots showing the infiltrating immune cells in DPN samples compared to normal samples. (**b**,**c**) The least absolute shrinkage and selection operator (LASSO) logistic regression algorithm was conducted to analyze the different infiltrates of immune cells in DPN and control samples. Different colors represent different types of immune cells. Trajectories of the independent variables of LASSO regression, the horizontal coordinates indicate the logarithm of the independent variable Lambda, and the vertical coordinates indicate the coefficients of the independent variables. LASSO regression under each Lambda confidence interval.

**Figure 7 biomolecules-13-00039-f007:**
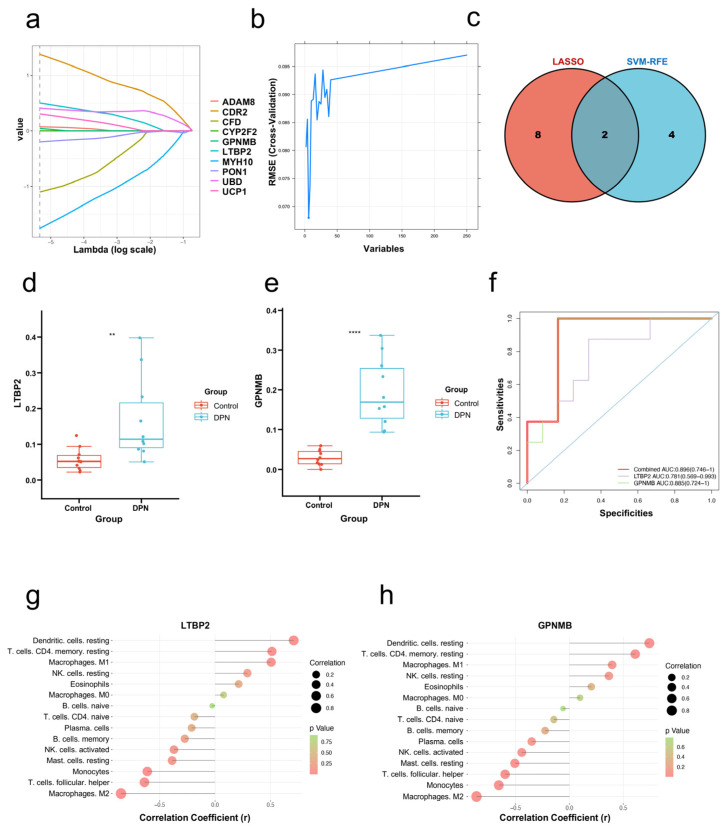
Screening and verification of biomarkers. (**a**) The LASSO logistic regression algorithm was used to screen biomarkers. Different colors represent different genes. (**b**) The Support vector machine-recursive feature elimination (SVM-RFE) algorithm was used to screen biomarkers. (**c**) Venn diagram for selecting identical potential biomarkers obtained by the two algorithms. (**d**,**e**) Detailed expression of two biomarkers in DPN and control groups in the validation dataset (** *p* < 0.01, **** *p* < 0.0001). (**f**) The diagnostic performance of selected biomarkers (AUC refers to the area under the ROC curve). (**g**) The correlation between differential immune infiltrating cells and LTBP2. (**h**) The correlation between differential immune infiltrating cells and GPNMB. The color of the dots represents the *p*-value and the size of the dots represents the strength of the correlation between genes and immune cells. A *p* value < 0.05 was considered statistically significant.

**Figure 8 biomolecules-13-00039-f008:**
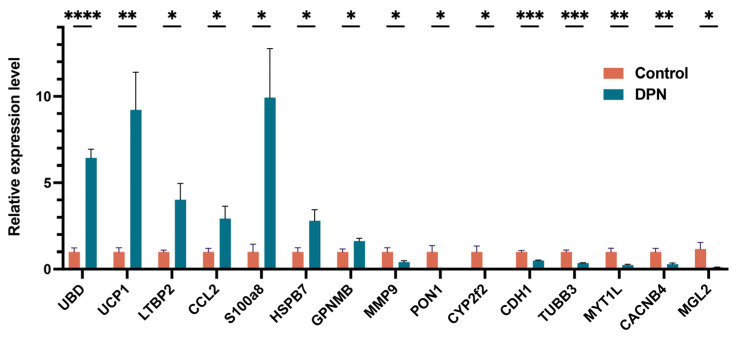
Confirmation of microarray data in sciatic nerves from db/db diabetic mice and db/+ nondiabetic mice by qRT-PCR. The microarray data and qRT-PCR results are consistent. Results were presented as means ± SEM of five independent experiments (* *p* < 0.05, ** *p* < 0.01, *** *p* < 0.001, **** *p* < 0.0001).

## Data Availability

The datasets generated during and/or analyzed during the current study are available in the Gene Expression Omnibus (GEO) datasets (http://www.ncbi.nlm.nih.gov/geo/, accessed on 10 March 2022).

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
