# Peer review of "Identification of Immune Infiltration and the Potential Biomarkers in Diabetic Peripheral Neuropathy through Bioinformatics and Machine Learning Methods"

_biomolecules, 2022, doi:10.3390/biom13010039_

Round 1

Reviewer 1 Report

Correctly report the meaning of the acronym DPN in the text.

Indicate references for the following statement :" However, traditional biomarkers have shortcomings and limitations, and few have been used clinically. Recently, molecules involved in several metabolic and signaling pathways associated 68 with DPN have been suggested as predictive biomarkers."

Change the writing  "(Figure 1a, b)" to (Figure 1a and 1b), change the writing "(Figure 1c, d)" to (Figure 1c and 1d) and so on for the other figures.

The paper is of interest, novel  and could have a scientific attraction. Data is well reported and, a part of some typo English mistakes, well written. I suggest to add to the title: a preliminary report, as reported by the Authors themselves in section conclusion.

Author Response

Those comments are all valuable and very helpful for revising and improving our paper, as well as the important guiding significance to our researches. We have studied comments carefully and have made correction which we hope meet with approval. The main corrections in the paper and the responds to the reviewer’s comments are as flowing:
1. Correctly report the meaning of the acronym DPN in the text.

Answer:

We added the meaning of the acronym DPN: diabetic peripheral neuropathy (DPN) in line 39 in section Introduction.

2. Indicate references for the following statement:" However, traditional biomarkers have shortcomings and limitations, and few have been used clinically. Recently, molecules involved in several metabolic and signaling pathways associated 68 with DPN have been suggested as predictive biomarkers."

Answer:

Following the reviewer’s comment, we have carefully consulted the relevant literatures and cited them in the revised manuscript.

  1. Marshall, A.; Alam, U.; Themistocleous, A.; Calcutt, N.; Marshall, A. Novel and Emerging Electrophysiological Biomarkers of Diabetic Neuropathy and Painful Diabetic Neuropathy. Clin. Ther. 2021, 43, 1441–1456, doi:10.1016/J.CLINTHERA.2021.03.020.
  2. Cho, N.R.; Yu, Y.; Oh, C.K.; Ko, D.S.; Myung, K.; Lee, Y.; Na, H.S.; Kim, Y.H. Neuropeptide Y: A Potential Theranostic Biomarker for Diabetic Peripheral Neuropathy in Patients with Type-2 Diabetes. Ther. Adv. Chronic Dis. 2021, 12, 1–11, doi:10.1177/20406223211041936.

3. Change the writing “(Figure 1a, b)" to (Figure 1a and 1b), change the writing "(Figure 1c, d)" to (Figure 1c and 1d) and so on for the other figures.

Answer:

We have changed all the writing like "(Figure 1a, b)" to (Figure 1a and 1b) in all figures.

Special thanks to you for your good comments.

Reviewer 2 Report

This study aimed at diabetic peripheral neuropathy related diagnostic markers.

1. Please demonstrated the identification of hub genes in details in the section 2.3.

2. The front size in Figure 3-7 is too small to read. Please make a modification.

3. In section 2.7, please list the function or pathway of those selected genes. Are they enriched in the top biomarker gene group?

Author Response

Those comments are all valuable and very helpful for revising and improving our paper, as well as the important guiding significance to our researches. We have studied comments carefully and have made correction which we hope meet with approval. The main corrections in the paper and the responds to the reviewer’s comments are as flowing:

  1. Please demonstrated the identification of hub genes in details in the section 2.3.

Answer:

We obtained the PPI network results of DEGs by STRING. Then 8 algorithms in the cytoHubba plugin including Stress, Radiality, MNC (Maximum Neighborhood Component), MCC (Maximal Clique Centrality),  EPC (Edge Percolated Component), EcCentricity, DMNC (Density of Maximum Neighborhood Component), Degree, Closeness, BottleNeck and Betweenness were used to calculated the score of each node gene. Taking the intersection of results of 8 algorithms, hub genes in PPI network were screened.

  1. The front size in Figure 3-7 is too small to read. Please make a modification.

Answer:

To facilitate reading, we have adjusted the front size in Figure 3-7.

  1. In section 2.7, please list the function or pathway of those selected genes. Are they enriched in the top biomarker gene group?

Answer:

In section 2.7, we selected DEGs with high fold change or high weight in the network to validate the analysis results. Furthermore, two biomarkers obtained by LASSO and SVM-RFE were also validated by qRT-PCR.

We have carefully queried these genes from the GeneCards website (https://www.genecards.org/) and list the function or pathway of these selected genes.

UBD (Ubiquitin D) is a Protein Coding gene. This gene encodes a protein which contains two ubiquitin-like domains and appears to have similar function to ubiquitin. Diseases associated with UBD include kidney disease and chronic kidney disease. Among its related pathways are Metabolism of proteins and BRCA1 Pathway.

Mitochondrial uncoupling proteins (UCP) are members of the family of mitochondrial anion carrier proteins (MACP). UCPs separate oxidative phosphorylation from ATP synthesis with energy dissipated as heat, also referred to as the mitochondrial proton leak. Diseases associated with UCP1 include Lipomatosis, Multiple Symmetric and Lipomatosis. Among its related pathways are "Respiratory electron transport, ATP synthesis by chemiosmotic coupling, and heat production by uncoupling proteins." and the fatty acid cycling model.

CCL2 is one of several cytokine genes clustered on the q-arm of chromosome 17. Chemokines are a superfamily of secreted proteins involved in immunoregulatory and inflammatory processes. Among its related pathways are MIF Mediated Glucocorticoid Regulation and TGF-Beta Pathway.

S100A8: The protein encoded by this gene is a member of the S100 family of proteins containing 2 EF-hand calcium-binding motifs. S100 proteins are localized in the cytoplasm and/or nucleus of a wide range of cells, and involved in the regulation of a number of cellular processes such as cell cycle progression and differentiation. S100A8 expression levels increased in many types of cancer, neurodegenerative disorders, inflammatory and autoimmune diseases and they are implicated in the numerous disease pathologies. Among its related pathways are diseases of immune system.

HSPB7: This gene encodes a small heat shock family B member that can heterodimerize with similar heat shock proteins. Diseases associated with HSPB7 include paraneoplastic polyneuropathy and distal hereditary motor type 2. Among its related pathways are differentiation of white and brown adipocyte.

MMP9: Proteins of the matrix metalloproteinase (MMP) family are involved in the breakdown of extracellular matrix in normal physiological processes, such as embryonic development, reproduction, and tissue remodeling, as well as in disease processes, such as arthritis and metastasis. Among its related pathways are Collagen chain trimerization and Matrix metalloproteinases.

PON1: This gene encodes a member of the paraoxonase family of enzymes and exhibits lactonase and ester hydrolase activity. Diseases associated with PON1 include microvascular complications of diabetes. Among its related pathways are Fatty acid metabolism and Drug ADME.

CYP2F2: This gene encodes a member of the cytochrome P450 superfamily of enzymes. Among its related pathways are Oxidation by cytochrome P450 and Metapathway biotransformation Phase I and II.

CDH1: This gene encodes a classical cadherin of the cadherin superfamily. Among its related pathways are Cell junction organization and Signaling by Rho GTPases.

TUBB3: This gene encodes a class III member of the beta tubulin protein family. This protein is primarily expressed in neurons and may be involved in neurogenesis and axon guidance and maintenance.

MYT1L: This gene encodes a member of the zinc finger superfamily of transcription factors whose expression, thus far, has been found only in neuronal tissues that function in the developing mammalian central nervous system.

CACNB4: This gene encodes a member of the beta subunit family of voltage-dependent calcium channel complex proteins. Among its related pathways are DREAM Repression and Dynorphin Expression and TCR Signaling (Qiagen).

Furthermore, we have altered section 2.7 in the revision and added discussion associated with some selected genes.

Special thanks to you for your good comments.